# Jointly Learning to Construct and Control Agents using Deep Reinforcement Learning

## Abstract

The physical design of a robot and the policy that controls its motion are inherently coupled. However, existing approaches largely ignore this coupling, instead choosing to alternate between separate design and control phases, which requires expert intuition throughout and risks convergence to suboptimal designs. In this work, we propose a method that jointly optimizes over the physical design of a robot and the corresponding control policy in a model-free fashion, without any need for expert supervision. Given an arbitrary robot morphology, our method maintains a distribution over the design parameters and uses reinforcement learning to train a neural network controller. Throughout training, we refine the robot distribution to maximize the expected reward. This results in an assignment to the robot parameters and neural network policy that are jointly optimal. We evaluate our approach in the context of legged locomotion, and demonstrate that it discovers novel robot designs and walking gaits for several different morphologies, achieving performance comparable to or better than that of hand-crafted designs.

## 1 Introduction

An agent's ability to navigate through and interact with its environment depends not just on its skill at planning and controlling its motion, but also on its physical design. Different physical designs are inherently better suited to different tasks and environments. By making appropriate choices during fabrication, mechanical elements can be designed to improve robustness to non-idealities such as errors in perception, delays in actuation, etc., and indeed, make control problem an easier one to solve. At the same time, robots that take different forms may find completely different control strategies to be optimal to complete the same task. Therefore, the physical and computational design of an agent are inherently coupled, and must ideally be jointly optimized if the robot is to successfully complete a task in a particular environment.

Consider the development of a legged robot for locomotion. Variations in physical design will require changes to the joint torques in order to preserve a particular locomotion behavior (e.g., a heavier torso requires greater torque at the ankle), and will likely result in completely different walking gaits, even when the morphology is preserved. In fact, some changes to design may render locomotion impossible for the target operating environment (e.g., a robot with long feet may be unable to locomote up an incline). Meanwhile, careful choice of bipedal design enables passive walking (McGeer, 1990; Goswami et al., 1998; Collins et al., 2001). It is therefore beneficial to not simply consider the robot's design or gait to be fixed, but to optimize both jointly for the target environment and task. Similar co-design can be beneficial in other settings—for example for the control policy and physical characteristics of digits in robotic grippers for grasping.

While a robot's physical design and the corresponding control policy are inherently coupled, most existing methods ignore this coupling, instead choosing to alternate between separate design and control phases. Existing approaches that jointly reason over design and control (Digumarti et al., 2014; Ha et al., 2017; Spielberg et al., 2017) assume knowledge of an accurate model of the robot dynamics and require expert supervision (e.g., to provide a suitable initial design and guide the optimization process). However, these restrictive assumptions limits their applicability to a handful of specific settings, and often yield solutions heavily influenced by expert intuition.

In this work, we seek a general approach—one that can optimize a robot's physical characteristics jointly with controllers of a desired complexity (Fig. 1), that can be applied to general tasks in some

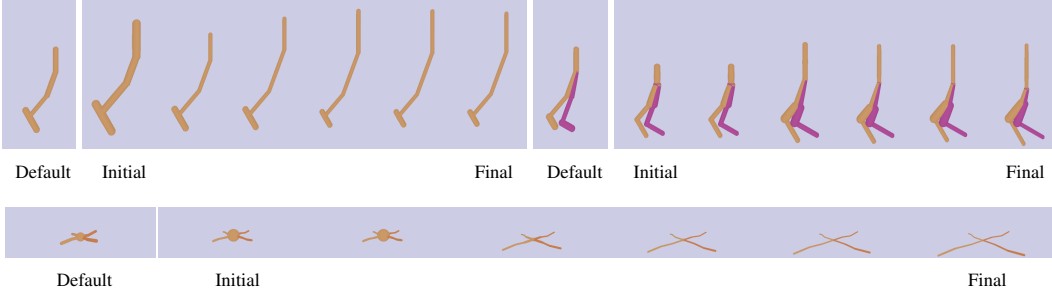

Figure 1: Our algorithm learns a robot's physical design jointly with the control policy. Here we show the learned designs evolving over time for the Hopper (top left), the Walker2d (top right) and the Ant (bottom), each with the default Roboschool design for comparison. Scale is fixed for each robot. Note that these designs correspond to modes of the distribution over robot designs that our algorithm maintains during training.

given environment, and that can explore the joint search space of physical design and computational control in a purely data-driven way, without a model of the robot dynamics and independent of the biases of expert intuition. We develop this approach in the context of determining the physical parameters of an articulated agent—the lengths and thicknesses of each limbs in a given morphology—through joint training with a neural network for control, with the objective of achieving locomotion. Our method maintains a distribution over these physical parameters, and simultaneously trains the parameters of this distribution with those of a neural network controller, using deep reinforcement learning. In this way, we pursue a design distribution and control policy that are jointly optimal for the given task and environment. Experimental results show that starting from random initializations, our approach is able to find novel designs and walking gaits that match or exceed the performance of manually designed agents. To the best of our knowledge, our method is the first to successfully carry out such a joint optimization of design and control in a completely model-free manner.

## 2 RELATED WORK

Attention has been paid recently to the problem of jointly optimizing the parameters that specify a robot's design and motion (e.g., gaits) or control. Early work in this area takes an evolutionary approach to optimizing a robot's design and controller, typically parameterized as a neural network, for virtual agents (Sims, 1994; Paul & Bongard, 2001; Agrawal et al., 2013) and physical robots (Lipson & Pollack, 2000; Bongard, 2011). Particularly relevant to our approach is the work of Ha et al. (2017), who relate design and motion parameters via a set of implicit functions that express robot dynamics, desired trajectories, and actuation limits. These functions encode a manifold that is then linearized to model the relationship between design and motion via the implicit function theorem. The method then solves for the desired parameters in a local fashion via constraint-based optimization. Similarly, Spielberg et al. (2017) describe an approach that jointly reasons over physical design and motion parameters for robots with articulated degrees of freedom (e.g., legged robots). They formulate the problem in the framework of trajectory optimization by incorporating parameters involved in the design of articulated robots, including dimensions, masses, mass distribution, and moments of inertia, together with the contact force and torque variables typically associated with trajectory optimization. They use weighted actuation cost as the objective, subject to a regularization penalty on parameters. Their method is guaranteed to find a feasible design so long as the problem is initialized within a small neighborhood of a feasible solution. Unlike our approach, their method requires that the user provide an estimate of the robot configuration at each time step, as well as an accurate analytic model of constraint dynamics (e.g., foot contact), which is computationally expensive to enforce. Geijtenbeek et al. (2013) propose a derivative-free strategy that optimizes over muscle routing and muscle-based control for simulated bipeds to realize a design that incorporates biomechanical constraints. Meanwhile, Digumarti et al. (2014) describe an evolutionary method that jointly reasons over design and motion parameters for legged robots, while also tuning the parameters of a robust controller that tracks these motions. Their method is limited to biologically

inspired quadrupedal foothold patterns, and does not account for contact dynamics. In contrast, our approach applies to arbitrary morphologies and does not require that we model contact dynamics.

While the focus on simultaneous optimization over robot design and motion parameters is relatively new, there is a long history of research focused on the related problem of co-design of physical structure and control (Reyer & Papalambros, 2002; Ravichandran et al., 2006). Park & Asada (1994) jointly optimize the link geometry of a high-speed robot arm along with the parameters of a PD joint controller. These methods often rely upon access to an accurate model of the robot dynamics in order to design the controller. In order to reduce the dependence upon a detailed analytical model, Pil & Asada (1996) use on-robot experiments to refine their model as part of the iterative design process. More recently, Villarreal-Cervantes et al. (2013) allow for some degree of uncertainty in the model by including the sensitivity of the design and control parameters to model uncertainty along with the task-specific optimization objectives. However, existing methods still rely upon access to an analytical model of the robot dynamics and are typically limited to simple (e.g., linear) control designs. Our method assumes only that the controller can be modeled via a convolutional neural network, and can thereby learn complex, highly nonlinear control policies with no a priori knowledge of the robot dynamics.

Far more attention has been paid to the individual problem of task-driven optimization of robot motion and control. Given an arbitrary robot design chosen by novice users, Megaro et al. (2015) describe an interactive design framework that solves for a realizable walking gait that results in stable locomotion. Similarly, Mordatch et al. (2012) synthesize emergent behaviors (motion) for arbitrary morphologies and tasks from high-level specifications, by jointly optimizing over contact and motion. Mordatch et al. (2015) build upon this work by training recurrent neural networks to serve as feedback controllers capable of producing complex, stable behaviors for a variety of dynamical systems. Their method interleaves supervised learning with trajectory optimization, and incorporates noise to improve generalizability. Meanwhile, there is a large body of literature that formulates motion synthesis as a trajectory optimization problem. This approach has proven effective at respecting contact constraints (e.g., between the foot and ground), which make controlling dynamic motion particularly challenging (Dai & Tedrake, 2016; Posa et al., 2016; Griffin & Grizzle, 2016). These approaches have been shown to generate sophisticated behaviors for complex robot designs (e.g., humanoids) (Tassa et al., 2012), and for robots of arbitrary morphologies using only a high-level specification of the robot's shape, gait, and task (Wampler et al., 2013). Related, a number of methods interleave trajectory optimization and supervised learning with neural network regression (Levine & Abbeel, 2014; Levine & Koltun, 2014; Mordatch & Todorov, 2014; Mordatch et al., 2015). Unlike our method, the use of trajectory optimization makes these approaches reliant upon knowledge of the model.

A great deal of attention of-late has focused on the problem of learning complex control policies directly from low-level sensory input, without any knowledge of the system dynamics. Methods that have proven particularly effective combine neural networks that learn representations of the high-dimensional raw sensor input with deep reinforcement learning (Riedmiller, 2005; Mnih et al., 2013; 2015; Schulman et al., 2015a). While much of the work in this area focuses on low-dimensional, discrete action spaces, several methods have been recently proposed that learn continuous control policies through deep reinforcement learning. These techniques have been applied to control simple, simulated robots (Wawrzyński, 2009; Wawrzyński & Tanwani, 2013; Watter et al., 2015; Lillicrap et al., 2015), perform robot manipulation (Levine et al., 2016a;b; Gu et al., 2017), and control legged robots (Schulman et al., 2015b; Peng et al., 2016).

Black-box optimization (Schmidhuber et al., 2007; Risi & Togelius, 2017) is an alternative to using reinforcement learning to training the control policy. These approaches have the advantage that they do not involve backpropagating gradients, are insensitive to reward sparsity, and can handle long time horizons. While black-box optimization strategies have traditionally been thought of as ill-suited to difficult reinforcement learning problems, Salimans et al. (2017) recently showed that they perform similarly to state-of-the-art RL methods on difficult continuous control problems, including those that involve locomotion. Closely related is the policy-gradient method of Sehnke et al. (2010), who define the policy as a distribution over the parameters of a controller, which they then sample over. This results in gradient estimates that are far less noisy than is typical of policy gradient algorithms. These approaches are similar to the way in which we learn the robot design, which we formulate as a Gaussian mixture model over design parameters. Indeed, the two referenced methods yield the same gradient estimate for Gaussian parameter distributions (Salimans et al., 2017).

Meanwhile, much work has focused on the problem of determining robot designs that meet the requirements of a particular task. Given a user demonstration of the desired behaviors, Coros et al. (2013) learn optimum kinematic linkages that are capable of reproducing these motions. Mehta et al. (2014) synthesize electromechanical robot designs in a compositional fashion based upon a complete user-specified structural specification of the robot. Mehta et al. (2016) build upon this work, allowing the user to specify functional objectives via structured English, which is parsed to a formal specification using linear temporal logic. Censi (2017) describes a theory for co-design that includes the ability to select discrete robot parts according to functional constraints, but do not reason over geometry or motion.

Related to our approach is recent work that jointly optimizes sensor design and inference algorithms for perception systems. Chakrabarti (2016) considers the problem of jointly learning a camera sensor's multiplexing pattern along with reconstruction methods for imaging tasks. They model inference as a neural network together and use stochastic gradient descent to backpropagate the loss to a neural layer representation of the multiplexing pattern. Related, Schaff et al. (2017) jointly learn design and inference for beacon-based localization. They encode beacon allocation (spatially and across transmission channels) as a differential neural layer that interfaces with a neural network for inference. Joint optimization then follows from standard techniques for training neural networks.

## 3 APPROACH

In this section, we begin by describing the standard reinforcement learning framework for training agent policies, and then describe how we extend this to also learn the physical design of the agent.

### 3.1 REINFORCEMENT LEARNING BACKGROUND

In the standard reinforcement learning setting, an agent interacts with its environment, usually a Markov Decision Process, over a number of discrete timesteps. At each time step $t$, the agent receives a state $s_t \in \mathcal{S}$ and takes action $a_t \in \mathcal{A}$ according to a policy $\pi : \mathcal{S} \to \mathcal{A}$. Then, the agent receives a scalar reward $r_t$ and the next state $s_{t+1}$ from the environment. This process continues until a terminal state is reached. The goal of reinforcement learning is to then find a policy $\pi^*$ that maximizes the expected return $\mathbb{E}[\mathcal{R}_t]$, where $\mathcal{R}_t = \sum_{i=0}^{\infty} \gamma^i r_{t+i}$ and $\gamma \in [0, 1)$ is a discount factor.

Policy gradient methods are a class of algorithms often used to optimize reinforcement learning problems, due to their ability to optimize cumulative reward and the ease with which the can be used with neural networks and other nonlinear function approximators. Consequently, they are commonly used for reinforcement learning problems that involve complex, continuous action spaces. Policy gradient methods directly parameterize a stochastic policy $\pi_\theta(a_t|s_t)$ and perform stochastic gradient ascent on the expected return. "Vanilla" policy gradient methods compute an estimate of the gradient $\nabla_\theta \mathbb{E}[\mathcal{R}_t]$ using a sample-based mean computed over $\nabla_\theta \log \pi_\theta(a_t|s_t)\mathcal{R}_t$ (Williams, 1992), which yields an unbiased gradient estimate (Sutton et al., 2000). While the variance in the resulting estimate decreases with the number of samples, sampling is computationally expensive. Another way to reduce variance while maintaining an unbiased estimate is to estimate the gradient by comparing the reward to a "baseline" reward $b(s_t)$.

These methods are effective but can be unstable, especially when used for deep reinforcement learning. Small changes to the policy parameters may cause large changes in the distribution of visited states. Several methods have been proposed to mitigate these effects. Among them, Schulman et al. (2015a) introduce Trust Region Policy Optimization (TRPO), which imposes a constraint on the KL-divergence between policies before and after an update. Recently, Schulman et al. (2017) proposed proximal policy optimization (PPO), a first-order class of methods similar to TRPO. PPO alternates between sampling data through interaction with the environment and optimizing the objective

$$\hat{\mathbb{E}}_t \left[ \min(r_t(\theta)\hat{A}_t, \mathrm{clip}(r_t(\theta), 1 - \epsilon, 1 + \epsilon)) \right], \tag{1}$$

where $r_t(\theta) = \frac{\pi_\theta(a_t|s_t)}{\pi_{\theta_{\mathrm{old}}}(a_t|s_t)}$ and $\hat{\mathbb{E}}_t$ represents an empirical average over a finite sample set. This objective seeks to maximize the expected return while encouraging a small step in policy space. The clipping within the objective removes any incentive for moving $r_t(\theta)$ outside the interval $[1-\epsilon, 1+\epsilon]$. The net effect of this objective is a simple, robust policy gradient algorithm that attains or matches state-of-the-art results on a wide array of tasks (Schulman et al., 2017).

---

**Algorithm 1** Joint Optimization

Initialize $\pi_\theta(a|s,r)$, $p_\phi(r)$
**while** True **do**
    **for** $i \in [1..n_p]$ **do**
        Sample $r_1, ..., r_n$ from $p_\phi$
        Collect rollouts with each design
        Update $\theta$ to maximize PPO's surrogate objective.
    **end for**
    **for** $i \in [1..n_r]$ **do**
        Sample $r_1, ..., r_m$ from $p_\phi$
        Compute returns $\mathcal{R}_1, ..., \mathcal{R}_m$.
        Update $\phi$ to maximize PPO's surrogate objective.
    **end for**
**end while**

---

### 3.2 JOINT OPTIMIZATION OF DESIGN AND CONTROL

We extend the standard reinforcement learning formulation by considering the space of possible robot designs $\Omega$. Specifically, we assume that for every design $\mathcal{E} \in \Omega$, we can define common state and action spaces $\mathcal{S}$ and $\mathcal{A}$, reward function $r(s,a)$, and initial state distribution $p_0(s)$, that are meaningful to all designs. The designs differ only in the transition dynamics they induce $p_\mathcal{E}(s'|s,a)$ and share a common action space $\mathcal{A}$—to achieve this, we assume a common morphology for all possible designs.

Our goal then is to find the optimal design $\mathcal{E}^*$ and policy $\pi^*_{\mathcal{E}^*}$ pair that maximizes the expected value of a given reward function. However, this is a non-linear, non-convex optimization problem over the spaces of all possible designs and all possible policies. Solving it exactly would require enumerating all possible designs (possibly through discretization of the design space), learning policies for each, and comparing the resulting expected rewards. This is computationally infeasible for all but the simplest of cases. Instead, we develop a gradient-based approach to solving this optimization problem with the following key components: (1) we maintain a multi-modal distribution over the space of physical designs and update this distribution using policy gradient methods in parameter space, similar to Sehnke et al. (2010) and Salimans et al. (2017); and (2) we train a single controller to act on all sampled designs during training, providing this controller with the design parameters of the specific sample it is controlling. We find that the multi-modal stochastic parameterization of the design space allows our method to explore the space more thoroughly (where a uni-modal Gaussian distribution would frequently get trapped in local minima). Moreover, a common controller makes optimization tractable, allowing efficient evaluation of unseen designs. We additionally benefit from learning common strategies across diverse designs, while still adapting different policies to different parts of the design space given the sample parameters as input. Together, these components enable efficient joint exploration of the design and policy spaces.

Formally, let $p(r;\phi)$ denote the distribution over designs, and $\pi(a_t|s_t,r;\theta)$ a stochastic control policy parameterized by $\phi$ and $\theta$ respectively. In our experiments, we use a neural network to model the control policy $\pi$, and a Gaussian mixture model as the parametric form of the distribution $p(r;\phi)$. Our goal is then to solve the following optimization problem:

$$\phi^*, \theta^* = \arg\max_{\phi,\theta} \mathbb{E}_{r,t}\left[\mathcal{R}_t\right], \qquad (2)$$

where $\mathbb{E}_{r,t}[\cdot]$ is the expectation over robots and trajectories those robots induce.

We use stochastic gradient-based updates to optimize Eqn. 2. Our method (Algorithm 3.2) alternates between updating the parameters of the policy and design distributions, $\theta$ and $\phi$, respectively. We empirically find this to yield convergence to better solutions in a reasonable number of iterations compared to performing simultaneous updates. We optimize the policy parameters $\theta$ using Proximal Policy Optimization. However, instead of collecting data from a single design, we sample a design $r$ after a fixed number $T$ of timesteps according to the distribution $p(r;\phi)$. After $n$ iterations of PPO, we freeze the policy parameters and proceed to optimize the parameters of the design distribution. Without knowledge of the model, we optimize the design parameters $\phi$ via policy gradient over

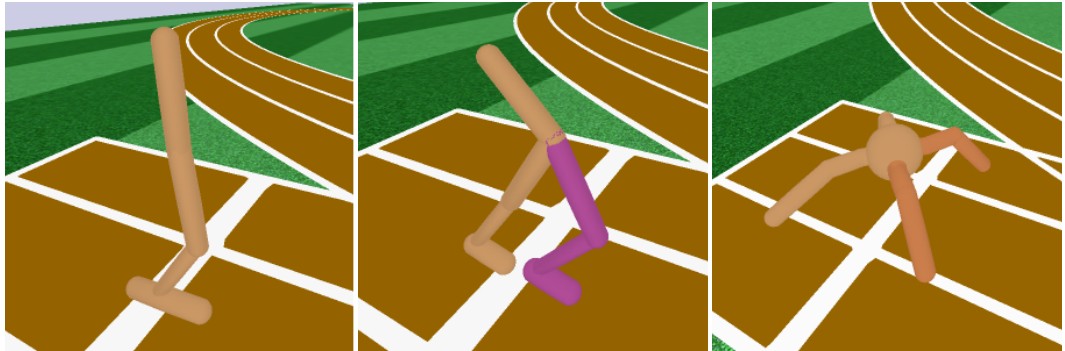

Figure 2: From left to right: the default Hopper, Walker2d, and Ant robots. The Hopper and Walker2d are constrained to walk in a line, while the Ant can walk anywhere in the plane.

parameter space, similar to Sehnke et al. (2010). This is equivalent to black-box optimization methods that have proven effective for complex learning problems (Salimans et al., 2017). We sample $m$ different designs and compute their returns for a single episode acting under policy $\pi(\cdot; \theta)$. We then use this data to shift the design distribution $p(\cdot; \phi)$ in a direction maximizing expected return. These iterations are repeated until convergence.

## 4 RESULTS

We validate our approach with three commonly used robots provided in OpenAI's Roboschool (Schulman et al., 2017). Environments in Roboschool are built on top of Bullet Physics, a popular open-source physics engine. In a series of locomotion experiments, we show that our approach not only discovers novel robot designs and gaits, but also outperforms two out of three of the hand-designed robots trained on the same task and yields performance comparable to the third.

### 4.1 EXPERIMENT DETAILS

We evaluate our approach on three legged locomotion tasks within Roboschool: RoboschoolHopper, RoboschoolWalker2d, and RoboschoolAnt (note that we subsequently drop the Roboschool prefix for brevity). Figure 2 depicts the default Roboschool design for each robot. These environments describe a locomotion task that has the robots moving along the positive x-axis (to the right in the figures) towards a distant goal. The reward function defined in these environments is a weighted sum of rewards for forward progress and staying upright, and penalties for applying torques and for joints that reach their rotational limits. The episode ends when the robot falls over or reaches a maximum number of timesteps.

We learn robot designs using the morphologies specified by the standard Roboschool designs. For each morphology, we parameterize the robot in terms of the length and radius (e.g., mass) of each link (or just the radius in the case of the sphere for the ant body). We impose symmetry, and share parameters across each leg for the Walker2d and Ant designs. This parameterization permits a wide variety of robots of different shapes and sizes, some of which are better suited to the locomotion objective than others, and some designs that do not permit a control policy that results in locomotion.

We model the control policy $\pi(a_t | s_t, \mathcal{E}; \theta_p)$ as a feed forward neural network consisting of three fully-connected layers with $64$ units and $\tanh$ activation functions. A final layer maps the output to the robot action space. With the exception of the last robot-specific layer, the architecture is the same for all experiments. Meanwhile, we represent the distribution over the parameterized robot design $p(\cdot; \theta_r)$ using a Gaussian mixture model with four mixture components (each with a diagonal covariance matrix over the different parameters). We initialize the means of each component randomly within a wide range, and initialize the variances in order to span the range of each parameter. We find that our approach maintains high variance distributions during early iterations—thereby continuing exploration of the design space—before committing to a chosen design. The appendix provides further details regardng the evolution of these distributions.

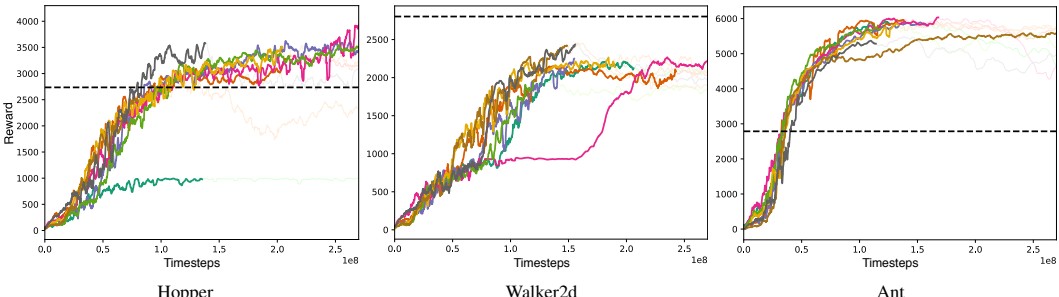

Figure 3: Training curves that show the evolution of reward across training iterations for the Hopper, Walker2d, and Ant environments over eight different random seeds. The black dashed line corresponds to the highest achievable performance of the corresponding baseline robot. In other figures, we display the best-performing checkpoint of each run. Therefore, we make each training curve transparent after it stops improving.

We train our method with eight threads of Proximal Policy Optimization for a total of 300M environment timesteps across all threads. We use the publicly available implementation of PPO provided in OpenAI Baselines (Hesse et al., 2017). We alternate between 50 policy PPO iterations and 2 design iterations, finding this ratio to yield convergence to good solutions. We set this and the other hyper-parameters on the Hopper and used the same settings for all experiments.

## 4.2 EXPERIMENTS

We evaluate the performance of our approach to joint optimization over robot design and control on the Hopper, Walker2d, and Ant robots, and compare against a policy trained for the standard Roboschool design. We evaluate the consistency and robustness of our approach relative to the initial robot parameter distribution using several different random seeds. For each robot morphology, we find that our method learns robot designs and the corresponding control policy that are either comparable (Walker2d) or exceed the performance (Hopper and Ant) achievable using the default designs (Fig. 3). We also see that our method achieves these levels of performance by discovering unique robot designs (Fig. 1) together with novel walking gaits (Fig. 4) that can not be achieved with the default designs. Note that our method was able to learn these designs and control policies from a random initialization, without access to a dynamics model or any expert supervision.

Our method learns a joint design and controller for the Hopper that outperforms the baseline by as much as 50%. Our learned robot exploits the small size of the foot to achieve faster, more precise walking gaits. Meanwhile, the longer torso of the learned robot improves stability, allowing it to maintain balance while locomoting at a faster pace. We found this behavior to be consistent across several different Hopper experiments, with the method converging to designs with a small foot and long, thin torso. In the appendix, we explore the stability of this design with respect to variations in the environment using the coefficient of friction as an example, and find the improvement to be consistent. For the ant, our optimization yields a physical design that is significantly different from the default design (Fig. 2). Consequently, the learned Ant drastically outperforms the baseline, improving reward by up to 116%. Our method learns a design with a small, lightweight body and extremely long legs. The long legs enable the ant to apply large torque at contact, allowing it to move at a fast pace.

Our framework learned different design-control pairs for the Walker2d that perform similarly to the default design. Across several different experiments, we see two distinct, successful designs and walking gaits. Interestingly, neither agent walks primarily on its feet. The first design has small, thick legs and long feet. The controller is then able to exploit the thick middle leg link, which protrudes slightly past the foot, to push off the ground. The long foot then provides additional balance. The second design is similar in geometry to the baseline Walker2d, but moves in a very different way. By lowering the knee joint and lengthening the foot, the Walker2d is able to efficiently balance on its knees and toes. This low stance allows the Walker2d to fully extend its leg backwards, creating a long, powerful stride, similar to that of a sprinter using starting blocks.

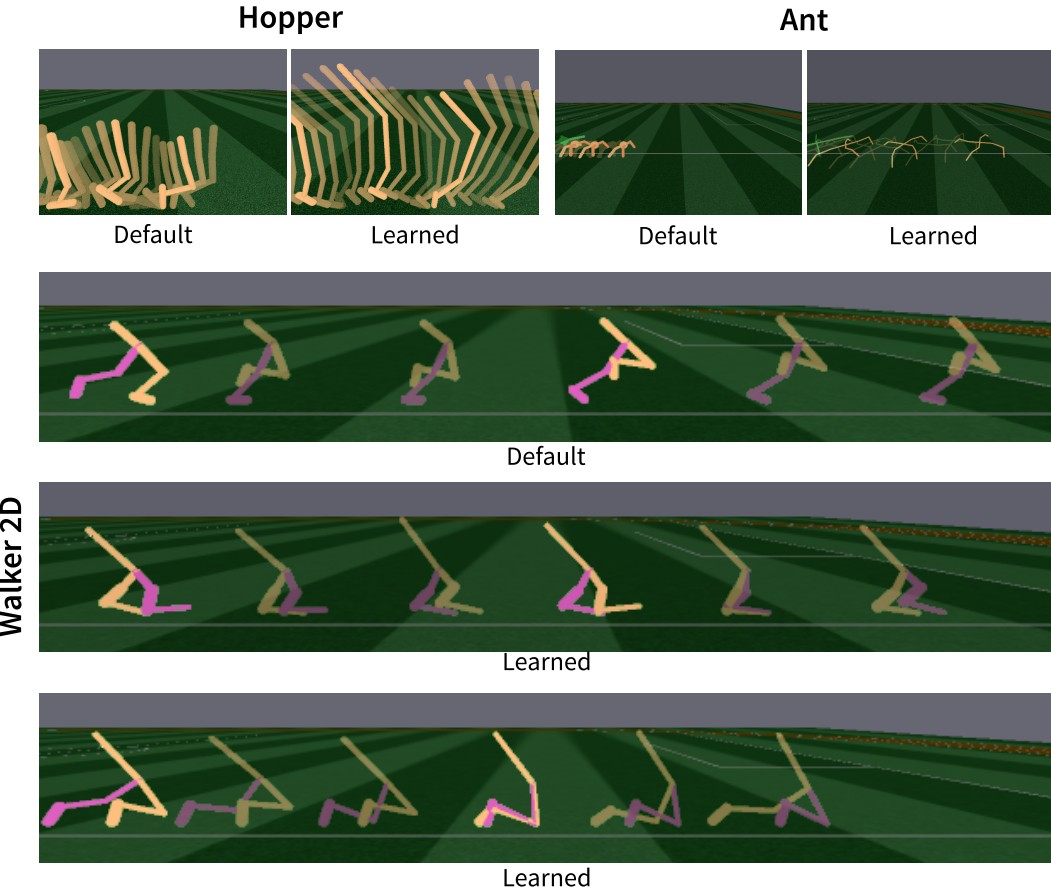

Figure 4: Here, we compare locomotion of default and learned robots, visualizing both their physical design and corresponding learned gaits. We pick the best results for Hopper and Ant, and two of the best results for Walker2d (due to diversity in gaits). Note that for each robot type, we show a blend of a fixed number of frames in the same time interval, allowing direct comparison between the speed with which different designs are able to locomote.

# 5 CONCLUSION

We proposed what is, to the best of our knowledge, the first model-free algorithm that jointly optimizes over the physical design of a robot and the corresponding control policy, without any need for expert supervision. Given an arbitrary morphology, our robot maintains a distribution over the robot design parameters and learns these parameters together with a neural network controller using policy gradient-based reinforcement learning. This results in an assignment to the policy over robot parameters and the control policy that are jointly optimal. We evaluated our approach on a series of different legged robot morphologies, demonstrating that it results in novel robot designs and walking gaits, achieving performance that either matches or exceeds that of manually defined designs.

Our findings suggest several avenues for future work. The most direct is extending the current approach to find optimized designs for uneven terrain, the presence of obstacles, changes in slope, variations in friction, etc. We are also interested in extending our framework to relax the assumption that the morphology is pre-defined. Finally, we are investigating applications to different types of agents and design spaces beyond legged robots (e.g., end-effectors), and exploring appropriate stochastic parameterization for such designs.

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

## 6  APPENDIX

The following provides further evaluation of our framework, including the behavior of the design distribution during training and the robustness of the learned designs to environment variations.

### 6.1  A - EVOLUTION OF GMM THROUGHOUT TRAINING

To provide insight into the training process of our algorithm, we evaluate the evolution of the Gaussian mixture model distribution throughout training of the best performing Hopper experiment. We initialize each component of the mixture model with random means and a diagonal covariance matrix chosen to cover the parameter space. The initial mixture weights are uniform. As shown in Figure 5 (left), roughly one third of the way through training, our algorithm converges to the most successful component. Additionally, we find that modes generally do not collapse (Figure 5 (right)). We find this behavior to be consistent across random seeds and different robots.

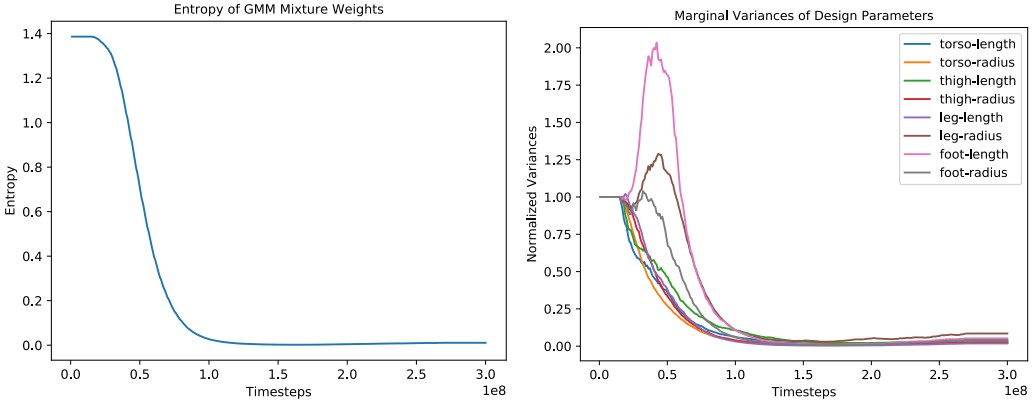

Figure 5: Plots of (left) the entropy of the mixture weights and (right) the marginal variance of each parameter throughout training. The curves are normalized such that all initial variances are one.

### 6.2  B - ROBUSTNESS OF LEARNED DESIGNS

It is often desirable for a robot to be able to operate in a variety of environments. In this section, we consider the robustness of a learned design to changes in friction. We conducted experiments on the Hopper in which we first learned the design and controller for one friction setting (0.8). We finetuned the controller in environments with different friction settings, while leaving the design fixed. We find the learned design to be reasonably robust with variability comparable to controllers finetuned for the default hand-crafted design (Fig. 6). Additionally, the learned design outperforms the hand-crafted one across the full range of friction values (although, for very low friction values, both designs essentially were unable to learn a successful gait).

Note that our framework can incorporate this goal of generalization by simply sampling from a diverse set of environments during training. But at the same time, it may be useful in some applications to seek out solutions that are specifically adapted to a relatively narrower set of environments, gaining better performance within this set at the cost of more general performance.

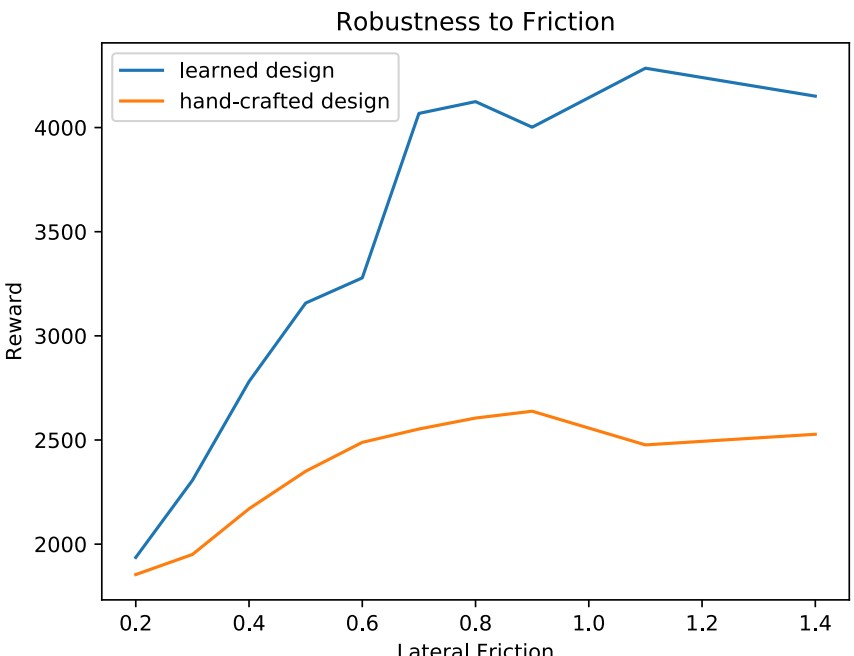

Figure 6: A visualization of accumulated reward for different friction values. The controllers and learned design were trained in an environment with a friction value of 0.8. We finetuned the controllers (but not the designs) for both the learned and hand-crafted designs for 10M timesteps for each friction value. Rewards are reported as an average over 100 episodes.

