# OpenReview forum: "Jointly Learning to Construct and Control Agents using Deep Reinforcement Learning"
_ICLR.cc/2018/Conference — Invite to Workshop Track_

### Official Review · AnonReviewer3 · 2017-11-26
**Paper of broad interest for control tasks**

**Rating:** 9
**Confidence:** 5

**Review:**

This is a well written paper, very nice work.
It makes progress on the problem of co-optimization of the physical parameters of a design
and its control system.  While it is not the first to explore this kind of direction,
the method is efficient for what it does; it shows that at least for some systems,
the physical parameters can be optimized without optimizing the controller for each
individual configuration. Instead, they require that the same controller works over an evolving
distribution of the agents.  This is a simple-but-solid insight that makes it possible
to make real progress on a difficult problem.

Pros:  simple idea with impact;  the problem being tackled is a difficult one
Cons:  not many;  real systems have constraints between physical dimensions and the forces/torques they can exert
       Some additional related work to consider citing.  The resulting solutions are not necessarily natural configurations,
   given the use of torques instead of musculotendon-modeling.  But the current system is a great start.

The introduction could also promote that over an evolutionary time-frame, the body and
control system (reflexes, muscle capabilities, etc.) presumably co-evolved.

The following papers all optimize over both the motion control and the physical configuration of the agents.
They all use derivative free optimization, and thus do not require detailed supervision or precise models
of the dynamics.

- Geijtenbeek, T., van de Panne, M., & van der Stappen, A. F. (2013). Flexible muscle-based locomotion
  for bipedal creatures. ACM Transactions on Graphics (TOG), 32(6), 206.
  (muscle routing parameters, including insertion and attachment points) are optimized along with the control).

- Sims, K. (1994, July). Evolving virtual creatures. In Proceedings of the 21st annual conference on
  Computer graphics and interactive techniques (pp. 15-22). ACM.
  (a combination of morphology, and control are co-optimized)

- Agrawal, S., Shen, S., & van de Panne, M. (2014). Diverse Motions and Character Shapes for Simulated
  Skills. IEEE transactions on visualization and computer graphics, 20(10), 1345-1355.
  (diversity in control and diversity in body morphology are explored for fixed tasks)

re: heavier feet requiring stronger ankles
This commment is worth revisiting.  Stronger ankles are more generally correlated with
a heavier body rather than heavy feet, given that a key role of the ankle is to be able
to provide a "push" to the body at the end of a stride, and perhaps less for "lifting the foot".

I am surprised that the optimization does not converge to more degenerate solutions
given that the capability to generate forces and torques is independent of the actual
link masses, whereas in nature, larger muscles (and therefore larger masses) would correlate
with the ability to generate larger forces and torques.  The work of Sims takes these kinds of
constraints loosely into account (see end of sec 3.3).

It would be interesting to compare to a baseline where the control systems are allowed to adapt to the individual design parameters.

I suspect that the reward function that penalizes torques in a uniform fashion across all joints would
favor body configurations that more evenly distribute the motion effort across all joints, in an effort
to avoid large torques.

Are the four mixture components over the robot parameters updated independently of each other
when the parameter-exploring policy gradients updates are applied?  It would be interesting
to know a bit more about how the mean and variances of these modes behave over time during
the optimization, i.e., do multiple modes end up converging to the same mean? What does the
evolution of the variances look like for the various modes?

---

> ### Author Response · Authors · 2017-12-20
> **Author Response**
>
> We are gratified by the reviewer's comments on the contributions of the paper, and thank you for the valuable feedback. Please find our responses below.
>
> RE: Relevant papers: Thank you for suggesting these papers. These are indeed relevant, and we will discuss them in the updated version.
>
> RE: Non-degenerate solutions: Without any constraints on the design space, we found that our method may converge to degenerate solutions. For example, without placing a lower bound on the length of each limb, the method exploits imperfections in the physics engine to learn a design and control strategy that achieve high reward, but are not realizable. We dealt with this by placing loose upper and lower bounds on the design parameters. We believe that similar application / manufacturing-specific constraints and costs---such as the correlation between actuator power and mass---can be easily incorporated into our framework.
>
> RE: Controller adaptation baseline: Unfortunately, it would be too computationally expensive to train/adapt separate controllers for individual designs when searching over a large enough design space. However, as we'll clarify in the paper, we're doing this already to some extent by providing the design parameters of a specific sampled instance as input to the controller, which can then learn to adapt its policy to that specific instance based on this input.
>
> RE: Distribution of applied torques: The reviewer is correct that applied torques are evenly distributed across all joints. This is likely because there is a squared penalty for applying torques at every joint. While this is a desirable property for real robots---reducing the stress on any particular part---it would be interesting to see what actuation would develop under other penalties, such as an L1 penalty.
>
> RE: Updates to mixture components: Our algorithm actually maintains a high-entropy distribution over the components till about a third of the way into training, before beginning to commit to a specific design (i.e., a single component, and eventually low variance within that component). Looking at the marginal variance of each parameter, we find that it also remains high early on in training---in fact, for some parameters like foot length and radius, it actually increases  before beginning to converge. The updated paper will provide visualizations of the evolution of these  distributions through training.

---

### Official Review · AnonReviewer2 · 2017-11-27
**Nice to see this topic pop up again, but paper is lacking comparisons and insights.**

**Rating:** 4
**Confidence:** 4

**Review:**

I'm glad to see the concept of jointly learning to control and evolve pop up again!

Unfortunately, this paper has a number of weak points that - I believe - make it unfit for publication in its current state.
Main weak points:
- No comparisons to other methods (e.g. switch between policy optimization for the controller and CMA-ES for the mechanical parameters). The basic result of the paper is that allowing PPO to optimize more parameters, achieves better results...
- One can argue that this is not true joint optimization Mechanical and control parameters are still treated differently. This begs the question: How should one define mechanical "variables" in order for them to behave similarly to other optimization variables (assuming that mechanical and control parameters influence the performance in a similar way)?

Additional relevant papers (slightly different approach):
http://www.pnas.org/content/108/4/1234.full#sec-1
http://ai2-s2-pdfs.s3.amazonaws.com/ad27/0104325010f54d1765fdced3af925ecbfeda.pdf

Minor issues:
Figure 1: please add labels/captions
Figure 2: please label the axes

---

> ### Author Response · Authors · 2017-12-20
> **Author Response**
>
> We thank the reviewer for their encouraging comments, and respond to specific points below:
>
> RE: Comparison to other methods: Note that our work seeks to enable automatic data-driven discovery of jointly optimal physical models and control policies. As part of this, we evaluate our method when it is initialized completely randomly---rather than with a "good" initial expert-guided guess. Thus, our experiments demonstrate the ability of our method to explore the entire design space, and potentially arrive at creative solutions far from expert intuition. As far as we know, all existing methods, including CMA-ES, require that the user decide on at least an initial parameterization, and conduct essentially a local search around a specific design. Therefore, these methods are not directly comparable. (Indeed, in our experiments we've found that initializing with a hand-crafted model---like the standard walker--- as the only component in our GMM allows our optimization to proceed quickly and improve that design. But this is not our goal.)
>
> RE: Alternating Optimization / just PPO over more parameters: Perhaps the biggest challenge we addressed in this paper relates to the design of an optimization strategy that is able to successfully search the joint design+control space, to arrive at good solutions while being computationally efficient. A key part of this is the alternating iterative procedure, which significantly improves computational efficiency and accelerates optimization.
>
> Moreover, note that we separate policy and design parameters in order to also allow a richer and more powerful parameterization of the network's belief distribution over good designs. We found that using a mixture model was key in allowing the optimization procedure to escape poor local optima, since in early iterations when the controller wasn't sufficiently sophisticated, it allowed the method to maintain a multi-modal distribution over a diverse set of possible "good" designs.
>
> We will expand on this in the updated version of the paper.
>
> RE: Relevant work: We sincerely thank the reviewer for these very relevant citations, and will discuss them in the updated version.
>
> Thank you for the review. We will address these points in the updated version.

---

### Official Review · AnonReviewer4 · 2017-12-06
**Well-written paper that could use additional results to show the method's merits and generality**

**Rating:** 5
**Confidence:** 3

**Review:**

The paper presents a model-free strategy for jointly optimizing robot design and a neural network-based controller. While it is well-written and covers quite a lot of related work, I have a few comments with regards to the algorithm and experiments.

- The algorithm boils down to an alternating policy gradient optimization of design and policy parameters, with policy parameters shared between all designs. This requires the policy to have to generalize across the current design distribution. How well the policy generalizes is then in turn fed back into the design parameter distribution, favoring those designs it could improve on the quickest. However, these designs are not guaranteed to be optimal in the long run, with further specialization. The results for the Walker2d might be hinting at this. A comparison between a completely shared policy vs. a specialized policy per design, possibly aided by a meta-learning technique to speed up the specialization, would greatly benefit the paper and motivate the use of a shared policy more quantitatively. If the condition of a common state/action space (morphology) is relaxed, then the assumption of smoothness in design space is definitely not guaranteed.
- Related to that, it would be interesting to see a visualization of the design space distribution. Is the GMM actually multimodal within a single run (which implies the policy is able to generalize across significantly different designs)?
- There are a separate number of optimization steps for the design and policy parameters within each iteration of the training loop, however the numbers used for the experiments are not listed.  It would be interesting to see what the influence of the ratio of these steps is, as well as to know how many design iterations were taken in order to get to those in Fig. 4. This is especially relevant if this technique is to be used with real physical systems. One could argue that, although not directly used for optimization or planning, the physics simulator acts a cheap dynamics model to test new designs.
- I wonder how robust and/or general the optimized designs are with respect to the task settings. Do small changes in the task or reward structure (i.e. friction or reward coefficients) result in wildly different designs? In practice, good robot designs are also robust and flexible and it would be great to get an idea how locally optimal the found designs are.

In summary, while the paper presents a simple but possibly effective and very general co-optimization procedure, the experiments and discussion don't definitively illustrate this.

Minor remarks:
- Sec. 1, final paragraph: "To do the best of our knowledge"
- Sec. 2, 3rd paragraph: "contract constraints"
- Sec. 4.1: a convolutional neural network consisting only of fully-connected layers can hardly be called convolutional
- Fig. 3: ±20% difference to the baseline for the Walker2d is borderline of what I would call comparable, but it seems like some of these runs have not converged yet so that difference might still decrease.
- Fig. 3: Please add x & y labels.

---

> ### Author Response · Authors · 2017-12-20
> **Author Response**
>
> We thank the reviewer for the valuable feedback, and respond to specific concerns below..
>
>
> RE: Local optima / shared policy: The reviewer is right in that the optimization may (and indeed likely does) converge to a local optima. But that is the fundamental challenge of what our method is trying to achieve: a joint search over design and policy space will have to involve, in all but the simplest cases, optimizing a complex, non-linear, and non-convex objective, at which point it is hard to guarantee convergence to a global optimum. (Indeed, even optimizing the control policy with a fixed design would not have such a guarantee). A major contribution of our work is in developing an optimization strategy that is able to find good solutions, if not globally optimal ones, with reasonable consistency, and we believe it constitutes an important step towards developing more efficient and successful optimization techniques for design+control problems.
>
> The shared policy actually ends up being critical to this effort. Firstly, we don't have any other option since it would simply be computationally infeasible to train a separate policy for every candidate design (which is also why we are unable to compare to such an approach---although we do compare to a policy learned with a fixed hand-crafted design as the baseline). However, we partially mitigate this by providing design parameters as input to the controller, allowing it to adapt its policy based on the specific design instance it is controlling. At the same time, having a common controller ensures that it is able to transfer knowledge of successful gaits and successful strategies between similar designs, and does not have to start training from scratch. This again is key in allowing optimization to succeed.
>
> We will update the paper to clarify this and expand the discussion of the motivation behind our design choices to provide the reader with greater intuition regarding the underlying optimization problem. We also agree that meta-learning would be an interesting approach to pursue in future work as a means to improve the efficiency of specialization.
>
> RE: Design space distribution: We find that the optimization process actually maintains a fairly high-variance multi-modal distribution over design choices till about a third of the way into training, before beginning to commit to a specific design. In most of the first 100M iterations, multiple components remain active, and the marginal variance of each physical parameter also remains high (indeed, for some parameters like foot length and radius, the variance actually increases first before beginning to converge). This exploration of the design space is in fact critical to successful optimization: we had initially attempted to use only a single Gaussian (i.e., just one component), which lead to greedy convergence to poor local optima.  We will update the paper to discuss this phenomenon, as well as visualize the evolution of the design parameter distribution.
>
> RE: Influence of optimization steps: We experimented with different alternation ratios between policy and design update iterations, which led to different speeds and qualities of convergence. We found that alternating too quickly results in the policy network not adapting fast enough to the changes in design parameters. If we alternated too slowly, we found that the algorithm takes a long time to converge or converges to poor local optima. We will include this discussion in the paper. All results in Figure 4 are reported after 300 million timesteps, which is roughly 5000 design iterations.
>
> RE: Robustness and generalizability: Based on the reviewer's comments, we conducted experiments on the hopper in which we fine-tuned the controller in environments with varying levels of friction, while keeping the learned design fixed. We found that the learned design was reasonably robust, and showed similar variability in performance compared to doing the same for the hand-crafted hopper---and the learned design outperformed the hand-crafted one across the full range of friction values (although, for very low friction values, both designs essentially were unable to learn a successful gait).
>
> Note that our framework can incorporate the goal of generalization by simply sampling from a diverse set of environment values during training. But at the same time, in some applications it may be useful to seek out solutions that are specifically adapted to a relatively narrower set of environment parameters, gaining better performance within this set at the cost of more general performance.

---

### Decision · Program_Chairs · 2018-01-29
**ICLR 2018 Conference Acceptance Decision**

**Decision:**

Invite to Workshop Track

**Comment:**

The chief contribution of this paper is to show that a single set of policy parameters can be optimized in an alternating fashion while the design parameters of the body are also optimized with policy gradients and sampled. The fact that this simple approach seems to work is interesting and worthy of note. However, the paper is otherwise quite limited - other methods are not considered or compared, incomplete experimental results are given, and important limitations of the method are not addressed. As it is an interesting but preliminary work, the workshop track would be appropriate.